# Learning Temporal Pose Estimation
# from Sparsely-Labeled Videos

**Gedas Bertasius**[1,2]**, Christoph Feichtenhofer**[1]**, Du Tran**[1]**, Jianbo Shi**[2]**, Lorenzo Torresani**[1]
[1]Facebook AI, [2]University of Pennsylvania

## Abstract

Modern approaches for multi-person pose estimation in video require large amounts of dense annotations. However, labeling every frame in a video is costly and labor intensive. To reduce the need for dense annotations, we propose a PoseWarper network that leverages training videos with sparse annotations (every $k$ frames) to learn to perform dense temporal pose propagation and estimation. Given a pair of video frames—a labeled Frame A and an unlabeled Frame B—we train our model to predict human pose in Frame A using the features from Frame B by means of deformable convolutions to implicitly learn the pose warping between A and B. We demonstrate that we can leverage our trained PoseWarper for several applications. First, at inference time we can reverse the application direction of our network in order to propagate pose information from manually annotated frames to unlabeled frames. This makes it possible to generate pose annotations for the entire video given only a few manually-labeled frames. Compared to modern label propagation methods based on optical flow, our warping mechanism is much more compact (6M vs 39M parameters), and also more accurate (88.7% mAP vs 83.8% mAP). We also show that we can improve the accuracy of a pose estimator by training it on an augmented dataset obtained by adding our propagated poses to the original manual labels. Lastly, we can use our PoseWarper to aggregate temporal pose information from neighboring frames during inference. This allows us to obtain state-of-the-art pose detection results on PoseTrack2017 and PoseTrack2018 datasets. Code has been made available at: `https://github.com/facebookresearch/PoseWarper`.

## 1 Introduction

In recent years, visual understanding methods [1–15] have made tremendous progress, partly because of advances in deep learning [16–19], and partly due to the introduction of large-scale annotated datasets [20, 21]. In this paper we consider the problem of pose estimation, which has greatly benefitted from the recent creation of large-scale datasets [22, 23]. Most of the recent advances in this area, though, have concentrated on the task of pose estimation in still-images [3, 23–27]. However, directly applying these image-level models to video is challenging due to nuisance factors such as motion blur, video defocus, and frequent pose occlusions. Additionally, the process of collecting annotated pose data in multi-person videos is costly and time consuming. A video typically contains hundreds of frames that need to be densely-labeled by human annotators. As a result, datasets for video pose estimation [22] are typically smaller and less diverse compared to their image counterparts [21]. This is problematic because modern deep models require large amounts of labeled data to achieve good performance. At the same time, videos have high informational redundancy as the content changes little from frame to frame. This raises the question of whether every single frame in a training video needs to be labeled in order to achieve good pose estimation accuracy.

To reduce the reliance on densely annotated video pose data, in this work, we introduce the Pose-Warper network, which operates on sparsely annotated videos, i.e., videos where pose annotations are given only every $k$ frames. Given a pair of frames from the same video—a labeled Frame A and an unlabeled Frame B—we train our model to detect pose in Frame A, using the features from Frame B. To achieve this goal, our model leverages deformable convolutions [28] across space and time. Through this mechanism, our model learns to sample features from an unlabeled Frame B to maximize pose detection accuracy in a labeled Frame A.

Our trained PoseWarper can then be used for several applications. First, we can leverage PoseWarper to propagate pose information from a few manually-labeled frames across the entire video. Compared to modern optical flow propagation methods such as FlowNet2 [29], our PoseWarper produces more accurate pose annotations ($88.7\%$ mAP vs $83.8\%$ mAP), while also employing a much more compact warping mechanism (6M vs 39M parameters). Furthermore, we show that our propagated poses can serve as effective pseudo labels for training a more accurate pose detector. Finally, our PoseWarper can be used to aggregate temporal pose information from neighboring frames during inference. This naturally renders the approach more robust to occlusion or motion blur in individual frames, and leads to state-of-the-art pose detection results on the PoseTrack2017 and PoseTrack2018 datasets [22].

## 2 Related Work

**Multi-Person Pose Detection in Images.** The traditional approaches for pose estimation leverage pictorial structures model [30–34], which represents human body as a tree-structured graph with pairwise potentials between the connected body parts. These approaches have been highly successful in the past, but they tend to fail if some of body parts are occluded. These issues have been partially addressed by the models that assume a non-tree graph structure [35–38]. However, most modern approaches for single image pose estimation are based on convolutional neural networks [3, 6, 23–27, 39–45]. The method in [3] regresses $(x, y)$ joint coordinates directly from the images. More recent work [25] instead predicts pose heatmaps, which leads to an easier optimization problem. Several approaches [24, 26, 39, 46] propose an iterative pose estimation pipeline where the predictions are refined at different stages inside a CNN or via a recurrent network. The methods in [6, 23, 45] tackle pose estimation problem in a top-down fashion, first detecting bounding boxes of people, and then predicting the pose heatmaps from the cropped images. The work in [24] proposes part affinity fields module that captures pairwise relationships between different body parts. The approaches in [42, 43] leverage a bottom-up pipeline first predicting the keypoints, and then assembling them into instances. Lastly, a recent work in [27], proposes an architecture that preserves high resolution feature maps, which is shown to be highly beneficial for the multi-person pose estimation task.

**Multi-Person Pose Detection in Video.** Due to a limited number of large scale benchmarks for video pose detection, there has been significantly fewer methods in the video domain. Several prior methods [22, 47, 48] tackle a video pose estimation task as a two-stage problem, first detecting the keypoints in individual frames, and then applying temporal smoothing techniques. The method in [49] proposes a spatiotemporal CRF, which is jointly optimized for the pose prediction in video. The work in [50] proposes a personalized video pose estimation framework, which is accomplished by finetuning the model on a few frames with high confidence keypoints in each video. The approaches in [51, 52] leverage flow based representations for aligning features temporally across multiple frames, and then using such aligned features for pose detection in individual frames.

In contrast to these prior methods, our primary objective is to learn an effective video pose detector from sparsely labeled videos. Our approach has similarities to the methods in [51, 52], which use flow representations for feature alignment. However, unlike our model, the methods in [51, 52] do not optimize their flow representations end-to-end with respect to the pose detection task. As we will show in our experiments, this is important for strong performance.

## 3 The PoseWarper Network

**Overview.** Our goal is to design a model that learns to detect pose from sparsely labeled videos. Specifically, we assume that pose annotations in training videos are available every $k$ frames. Inspired by a recent self-supervised approach for learning facial attribute embeddings [53], we formulate the following task. Given two video frames—a labeled Frame A and an unlabeled Frame B—our model

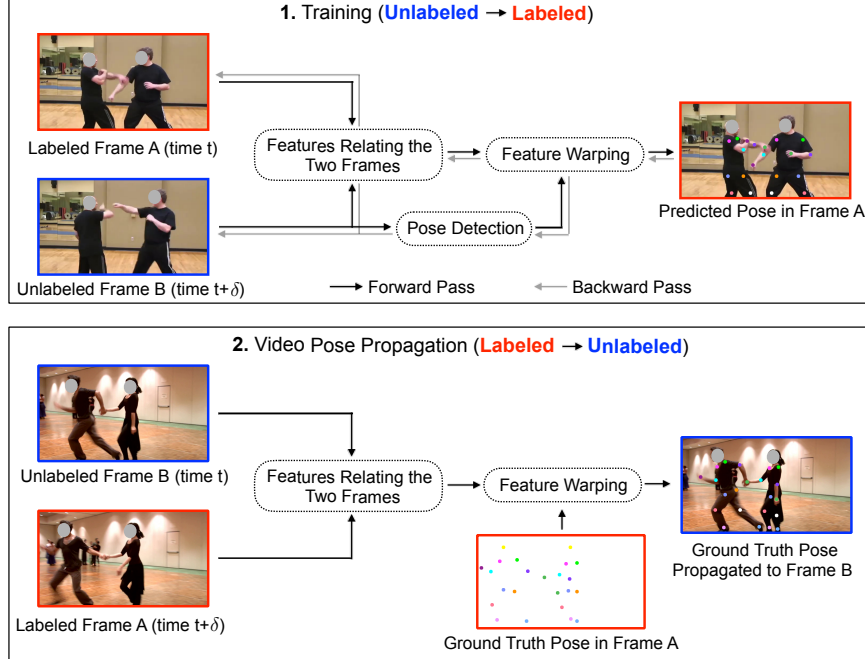

Figure 1: A high level overview of our approach for using sparsely labeled videos for pose detection. Faces in the figure are artificially masked for privacy reasons. In each training video, pose annotations are available only every $k$ frames. During training, our system considers a pair of frames–a labeled Frame A, and an unlabeled Frame B, and aims to detect pose in Frame A, using the features from Frame B. Our training procedure is designed to achieve two goals: 1) our model must be able to extract motion offsets relating these two frames. 2) Using these motion offsets our model must then be able to rewarp the detected pose heatmap extracted from an unlabeled Frame B in order to optimize the accuracy of pose detection in a labeled Frame A. After training, we can apply our model in reverse order to propagate pose information across the entire video from ground truth poses given for only a few frames.

is allowed to compare Frame A to Frame B but it must predict Pose A (i.e., the pose in Frame A) using the features from Frame B, as illustrated in Figure 1 (top).

At first glance, this task may look overly challenging: how can we predict Pose A by merely using features from Frame B? However, suppose that we had body joint correspondences between Frame A and Frame B. In such a scenario, this task would become trivial, as we would simply need to spatially "warp" the feature maps computed from frame B according to the set of correspondences relating frame B to frame A. Based on this intuition, we design a learning scheme that achieves two goals: 1) By comparing Frame A and Frame B, our model must be able to extract motion offsets relating these two frames. 2) Using these motion offsets our model must be able to rewarp the pose extracted from an unlabeled Frame B in order to optimize pose detection accuracy in a labeled Frame A.

To achieve these goals, we first feed both frames through a backbone CNN that predicts pose heatmaps for each of the frames. Then, the resulting heatmaps from both frames are used to determine which points from Frame B should be sampled for detection in Frame A. Finally, the resampled pose heatmap from Frame B is used to maximize accuracy of Pose A.

**Backbone Network.** Due to its high efficiency and accuracy, we use the state-of-the-art High Resolution Network (HRNet-W48) [27] as our backbone CNN. However, we note that our system can easily integrate other architectures as well. Thus, we envision that future improvements in still-image pose estimation will further improve the effectiveness of our approach.

**Deformable Warping.** Initially, we feed Frame A and Frame B through our backbone CNN, which outputs pose heatmaps $f_A$ and $f_B$. Then, we compute the difference $\psi_{A,B} = f_A - f_B$. The resulting feature tensor $\psi_{A,B}$ is provided as input to a stack of $3 \times 3$ simple residual blocks (as in standard ResNet-18 or ResNet-34 models), which output a feature tensor $\phi_{A,B}$. The feature tensor $\phi_{A,B}$ is then fed into five $3 \times 3$ convolutional layers each using a different dilation rate $d \in \{3, 6, 12, 18, 24\}$

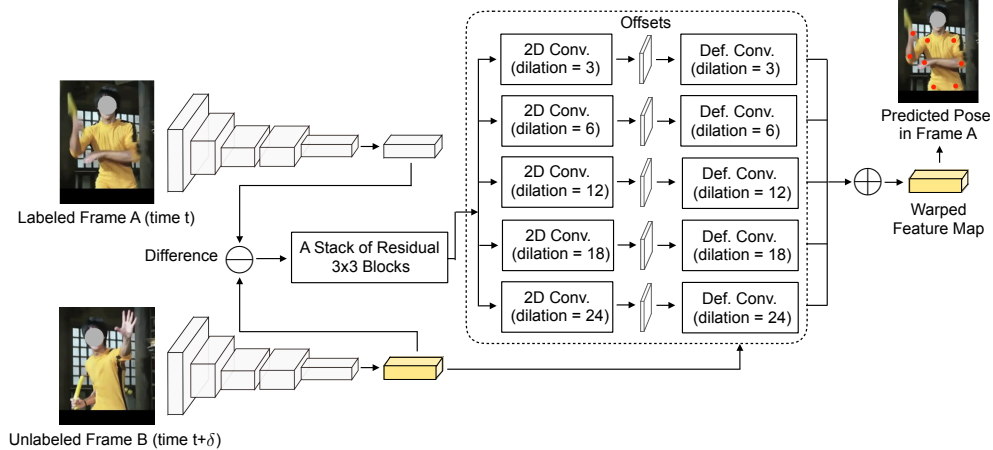

Figure 2: An illustration of our PoseWarper architecture. Given a labeled Frame A and an unlabeled Frame B, which are separated by $\delta$ steps in time, our goal is to detect pose in a labeled Frame A using the features from an unlabeled Frame B. First, we predict pose heatmaps for both frames. Then, we compute the difference between pose heatmaps in Frame A and Frame B and feed it through a stack of $3 \times 3$ residual blocks. Afterwards, we attach five $3 \times 3$ convolutional layers with dilation rates $d \in \{3, 6, 12, 18, 24\}$ and predict five sets of offsets $o^{(d)}(p_n)$ for each pixel location $p_n$. The predicted offsets are used to rewarp pose heatmap B. All five rewarped heatmaps are then summed and the resulting tensor is used to predict pose in Frame A.

to predict five sets of offsets $o^{(d)}(p_n)$ at all pixel locations $p_n$. The motivation for using different dilation rates at the offset prediction stage comes from the need to consider motion cues at different spatial scales. When the body motion is small, a smaller dilation rate may be more useful as it captures subtle motion cues. Conversely, if the body motion is large, using large dilation rate allows us to incorporate relevant information further away. Next, the predicted offsets are used to spatially rewarp the pose heatmap $f_B$. We do this for each of the five sets of offsets $o^{(d)}$, and then sum up all five rewarped pose heatmaps to obtain a final output $g_{A,B}$, which is used to predict pose in Frame A.

We implement the warping mechanism via a deformable convolution [28], which takes 1) the offsets $o^{(d)}(p_n)$, and 2) the pose heatmap $f_B$ as its inputs, and then outputs a newly sampled pose heatmap $g_{A,B}$. The subscript $(A, B)$ is used to indicate that even though $g_{A,B}$ was resampled from tensor $f_B$, the offsets for rewarping were computed using $\psi_{A,B}$, which contains information from both frames. An illustration of our architecture is presented in Figure 2.

**Loss Function.** As in [27], we use a standard pose estimation loss function which computes a mean squared error between the predicted, and the ground-truth heatmaps. The ground-truth heatmap is generated by applying a 2D Gaussian around the location of each joint.

**Pose Annotation Propagation.** During training, we force our model to warp pose heatmap $f_B$ from an unlabeled frame B such that it would match the ground-truth pose heatmap in a labeled Frame A. Afterwards, we can reverse the application direction of our network. This then, allows us to propagate pose information from manually annotated frames to unlabeled frames (i.e. from a labeled Frame A to an unlabeled Frame B). Specifically, given a pose annotation in Frame A, we can generate its respective ground-truth heatmap $y_A$ by applying a 2D Gaussian around the location of each joint (identically to how it was done in [23, 27]. Then, we can predict the offsets for warping ground-truth heatmap $y_A$ to an unlabeled Frame B, from the feature difference $\psi_{B,A} = f_B - f_A$. Lastly, we use our deformable warping scheme to warp the ground-truth pose heatmap $y_A$ to Frame B, thus, propagating pose annotations to unlabeled frames in the same video. See Figure 1 (bottom) for a high-level illustration of this scheme.

**Temporal Pose Aggregation at Inference Time.** Instead of using our model to propagate pose annotations on training videos, we can also use our deformable warping mechanism to aggregate pose information from nearby frames during inference in order to improve the accuracy of pose detection. For every frame at time $t$, we want to aggregate information from all frames at times $t + \delta$ where $\delta \in \{-3, -2, -1, 0, 1, 2, 3\}$. Such a pose aggregation procedure renders pose estimation more robust to occlusions, motion blur, and video defocus.

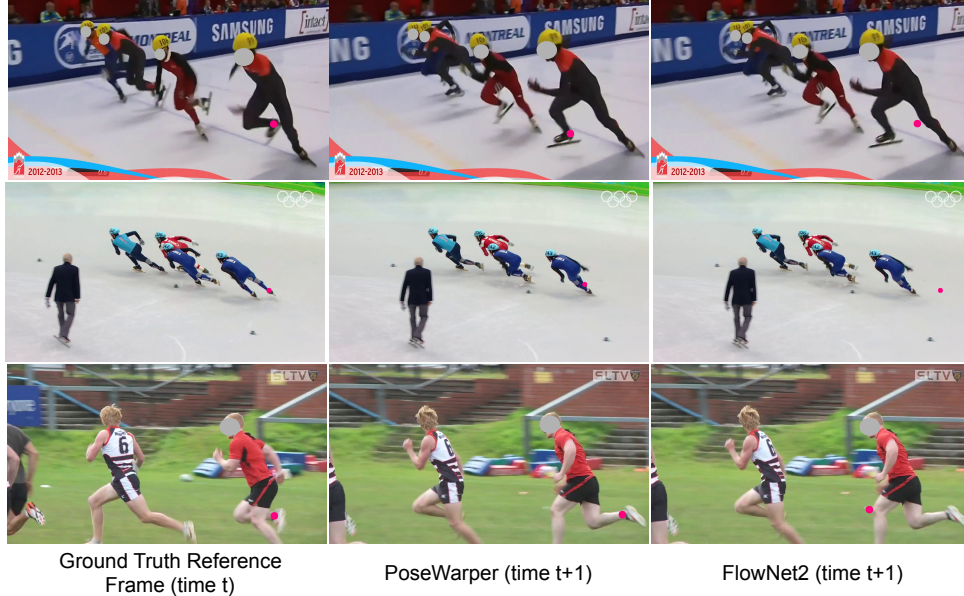

| Ground Truth Reference<br>Frame (time t) | PoseWarper (time t+1) | FlowNet2 (time t+1) |

Figure 3: The results of a video pose propagation task by our PoseWarper and FlowNet2 [29]. The first frame in each 3-frame sequence illustrates a *labeled* reference frame at time t. For simplicity, we show only the "right ankle" body joint for one person, denoted by a **pink** circle in each of the frames (please zoom in for a clearer view). The second frame depicts our propagated "right ankle" detection from the labeled frame in time t to the unlabeled frame in time t+1. The third frame shows the propagated detection in frame t+1 produced by the FlowNet2 baseline. In contrast to our method, FlowNet2 fails to propagate poses when there is large motion, blurriness or occlusions.

Consider a pair of frames, $I_t$ and $I_{t+\delta}$. In this case, we want to use pose information from frame $I_{t+\delta}$ to improve pose detection in frame $I_t$. To do this, we first feed both frames through our trained PoseWarper model, and obtain a spatially rewarped (resampled) pose heatmap $g_{t,t+\delta}$, which is aligned with respect to frame $I_t$ using the features from frame $I_{t+\delta}$. We can repeat this procedure for every $\delta$ value, and then aggregate pose information from multiple frames via a summation as $\sum_\delta g_{t,t+\delta}$.

**Implementation Details.** Following the framework in [27], for training, we crop a $384 \times 288$ bounding box around each person and use it as input to our model. During training, we use ground truth person bounding boxes. We also employ random rotations, scaling, and horizontal flipping to augment the data. To learn the network, we use the Adam optimizer [54] with a base learning rate of $10^{-4}$, which is reduced to $10^{-5}$ and $10^{-6}$ after 10, and 15 epochs, respectively. The training is performed using 4 Tesla M40 GPUs, and is terminated after 20 epochs. We initialize our model with a HRNet-W48 [27] pretrained for a COCO keypoint estimation task. To train the deformable warping module, we select Frame B, with a random time-gap $\delta \in [-3, 3]$ relative to Frame A. To compute features relating the two frames, we use twenty $3 \times 3$ residual blocks each with 128 channels. Even though this seems like many convolutional layers, due to a small number of channels in each layer, this amounts to only 5.8M parameters (compared to 39M required to compute optical flow in [29]). To compute the offsets $o^{(d)}$, we use five $3 \times 3$ convolutional layers, each using a different dilation rate ($d = 3, 6, 12, 18, 24$). To resample the pose heatmap $f_B$, we employ five $3 \times 3$ deformable convolutional layers, each applied to one of the five predicted offset maps $o^{(d)}$. The five deformable convolution layers too employ different dilation rates of $3, 6, 12, 18, 24$. During testing, we follow the same two-stage framework used in [27, 23]: first, we detect the bounding boxes for each person in the image using the detector in [48], and then feed the cropped images to our pose estimation model.

## 4 Experiments

In this section, we present our results on the PoseTrack [22] dataset. We demonstrate the effectiveness of our approach on three applications: 1) video pose propagation, 2) training a network on annotations augmented with propagated pose pseudo-labels, 3) temporal pose aggregation during inference.

Table 1: The results of video pose propagation on the PoseTrack2017 [22] validation set (measured in mAP). We propagate pose information across the entire video from the manual annotations provided in few frames. To study the effect of different levels of dilated convolutions in our PoseWarper architecture, we also include several ablation baselines (see the bottom half of the table).

| Method | Head | Shoulder | Elbow | Wrist | Hip | Knee | Ankle | Mean |
|---|---|---|---|---|---|---|---|---|
| Pseudo-labeling w/ HRNet [27] | 79.1 | 86.5 | 81.4 | 74.7 | 81.4 | 79.4 | 72.3 | 79.3 |
| Optical Flow Propagation (Farneback [55]) | 76.5 | 82.3 | 74.3 | 69.2 | 80.8 | 74.8 | 70.1 | 75.5 |
| Optical Flow Propagation (FlowNet2 [29]) | 82.7 | 91.0 | 83.8 | 78.4 | 89.7 | 83.6 | 78.1 | 83.8 |
| PoseWarper (no dilated convs) | 86.1 | 91.7 | 88.0 | 83.5 | 90.2 | 87.3 | 84.6 | 87.2 |
| PoseWarper (1 dilated conv) | 85.0 | 91.6 | 88.0 | 83.7 | 89.6 | 87.3 | 84.7 | 87.0 |
| PoseWarper (2 dilated convs) | 85.8 | 92.4 | 88.8 | 84.9 | 91.0 | 88.4 | 86.0 | 88.0 |
| PoseWarper (3 dilated convs) | 86.1 | 92.6 | 89.2 | 85.5 | 91.3 | 88.8 | 86.3 | 88.4 |
| PoseWarper (4 dilated convs) | **86.3** | 92.6 | **89.5** | 85.9 | **91.9** | 88.8 | 86.4 | 88.6 |
| PoseWarper (5 dilated convs) | 86.0 | **92.7** | **89.5** | **86.0** | 91.5 | **89.1** | **86.6** | **88.7** |

## 4.1 Video Pose Propagation

**Quantitative Results.** To verify that our model learns to capture pose correspondences, we apply it to the task of video pose propagation, i.e., propagating poses across time from a few labeled frames. Initially, we train our PoseWarper in a sparsely labeled video setting according to the procedure described above. In this setting, every $7^{th}$ frame of a training video is labeled, i.e. there are 6 unlabeled frames between each pair of manually labeled frames. Since each video contains on average 30 frames, we have approximately 5 annotated frames uniformly spaced out in each video. Our goal then, is to use our learned PoseWarper to propagate pose annotations from manually-labeled frames to all unlabeled frames in the same video. Specifically, for each labeled frame in a video, we propagate its pose information to the three preceding and three subsequent frames. We train our PoseWarper on sparsely labeled videos from the training set of PoseTrack2017 [22] and then perform our evaluations on the validation set.

To evaluate the effectiveness of our approach, we compare our model to several relevant baselines. As our weakest baseline, we use our trained HRNet [27] model that simply predicts pose for every single frame in a video. Furthermore, we also include a few propagation baselines based on warping annotations using optical flow. The first of these uses a standard Farneback optical flow [55] to warp the manually-labeled pose in each labeled frame to its three preceding and three subsequent frames. We also include a more advanced optical flow propagation baseline that uses FlowNet2 optical flow [29]. Finally, we evaluate our PoseWarper model.

In Table 1, we present our quantitative results for video pose propagation. The evaluation is done using an mAP metric as in [42]. Our best model achieves a $88.7\%$ mAP, while the optical flow propagation baseline using FlowNet2 [29] yields an accuracy of $83.8\%$ mAP. We also note that compared to the FlowNet2 [29] propagation baseline, our PoseWarper warping mechanism is not only more accurate, but also significantly more compact (6M vs 39M parameters).

**Ablation Studies on Dilated Convolution.** In Table 1, we also present the results investigating the effect of different levels of dilated convolutions in our PoseWarper architecture. We evaluate all these variants on the task of video pose propagation. First, we report that removing dilated convolution blocks from the original architecture reduces the accuracy from $88.7$ mAP to $87.2$ mAP. We also note that a network with a single dilated convolution (using a dilation rate of 3) yields $87.0$ mAP. Adding a second dilated convolution level (using dilation rates of $3, 6$) improves the accuracy to $88.0$. Three dilation levels (with dilation rates of $3, 6, 12$) yield a mAP of $88.4$ and four levels (dilation rates of $3, 6, 12, 18$) give a mAP of $88.6$. A network with $5$ dilated convolution levels yields $88.7$ mAP. Adding more dilated convolutions does not improve the performance further. Additionally, we also experimented with two networks that use dilation rates of $1, 2, 3, 4, 5$, and $4, 8, 16, 24, 32$, and report that such models yield mAPs of $88.6$ and $88.5$, respectively, which are slightly lower.

**Qualitative Comparison to FlowNet2.** In Figure 3, we include an illustration of the motion encoded by PoseWarper, and compare it to the optical flow computed by FlowNet2 for the video pose propagation task. The first frame in each 3-frame sequence illustrates a *labeled* reference frame at time t. For a cleaner visualization, we show only the "right ankle" body joint for one person, which is marked with a **pink** circle in each of the frames. The second frame depicts our propagated "right

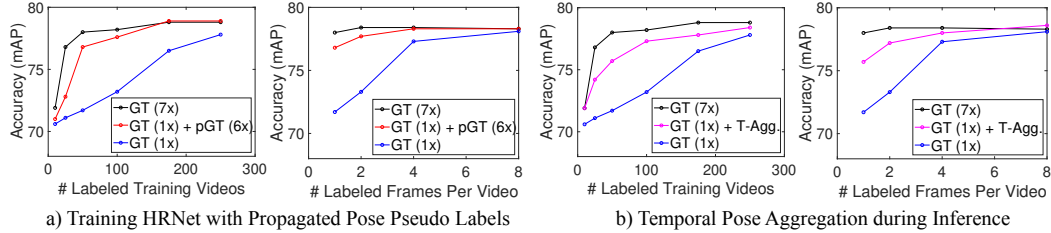

a) Training HRNet with Propagated Pose Pseudo Labels    b) Temporal Pose Aggregation during Inference

Figure 4: A figure illustrating the value of a) training a standard HRNet [27] using our propagated pose pseudo labels (left), and b) our temporal pose aggregation scheme during inference. In both settings, we study pose detection performance as a function of 1) number of sparsely-labeled training videos (with 1 manually-labeled frame per video), and 2) number of labeled frames per video (with 50 sparsely-labeled videos in total). All baselines are based on retraining the standard HRNet [27] model on the different training sets. The "GT (1x)" baseline is trained in a standard way on sparsely labeled video data. The "GT (7x)" baseline uses 7x more manually annotated data relative to the "GT (1x)" baseline. Our approach on the left subfigure ("GT (1x) + pGT (6x)"), augments the original sparsely labeled video data with our propagated pose pseudo labels (6 nearby frames for every manually-labeled frame). Lastly, in b) "GT (1x) + T-Agg" denotes the use of PoseWarper to fuse pose information from multiple neighboring frames during inference (training is done as in "GT (1x)" baseline). From the results, we observe that both application modalities of PoseWarper provide an effective way to achieve strong pose accuracy while reducing the number of manual annotations.

ankle" detection from the labeled frame in time t to the unlabeled frame in time t+1. The third frame shows the propagated detection in frame t+1 produced by the FlowNet2 baseline. These results suggest that FlowNet2 struggles to accurately warp poses if 1) there is large motion, 2) occlusions, or 3) blurriness. In contrast, our PoseWarper handles these cases robustly, which is also indicated by our results in Table 1 (i.e., 88.7 vs 83.8 mAP w.r.t. FlowNet2).

## 4.2 Data Augmentation with PoseWarper

Here we consider the task of propagating poses on sparsely labeled training videos using PoseWarper, and then using them as pseudo-ground truth labels (in addition to the original manual labels) to train a standard HRNet-W48 [27]. For this experiment, we study the pose detection accuracy as a function of two variables: 1) the total number of sparsely-labeled videos, and 2) the number of manually-annotated frames per video. We aim to study how much we can reduce manual labeling through our mechanism of pose propagation, while maintaining strong pose accuracy. Note, that we first train our PoseWarper on sparsely labeled videos from the training set of PoseTrack2017 [22]. Then, we propagate pose annotations on the same set of training videos. Afterwards, we retrain the model on the joint training set comprised of sparse manual pose annotations and our propagated poses. Lastly, we evaluate this trained model on the validation set.

All results are based on a standard HRNet [27] model trained on different forms of training data. "GT (1x)" refers to a model trained on sparsely labeled videos using ground-truth annotations only. "GT (7x)" baseline employs 7x more manually-annotated poses relative to "GT (1x)" (the annotations are part of the PoseTrack2017 training set). In comparison, our approach ("GT (1x) + pGT (6x)"), is trained on a joint training set consisting of sparse manual pose annotations (same as "GT (1x)" baseline) and our propagated poses (on the training set of PoseTrack2017), which we use as pseudo ground truth data (pGT). As before, for every labeled frame we propagate the ground truth pose to the 3 previous and the 3 subsequent frames, which allows us to expand the training set by 7 times.

Based on the results in the left subfigure of Figure 4, we can draw several conclusions. First, we note that when there are very few labeled videos (i.e., 5), all three baselines perform poorly (leftmost figure). This indicates that in this setting there is not enough data to learn an effective pose detection model. Second, we observe that when the number of labeled videos is somewhat reasonable (e.g., $50 - 100$), our approach significantly outperforms the "GT (1x)" baseline, and is only slightly worse relative to the "GT (7x)" baseline. As we increase the number of labeled videos, the gaps among the three methods shrink, suggesting that the model becomes saturated.

Table 2: Multi-person pose estimation results on the validation and test sets of PoseTrack2017 and PoseTrack2018 datasets. Even though our model is designed to improve pose detection in scenarios involving sparsely-labeled videos, here we show that our temporal pose aggregation scheme during inference is also useful for models trained on densely labeled videos. We improve upon the state-of-the-art single-frame baselines [23, 27, 56].

| Dataset | Method | Head | Shoulder | Elbow | Wrist | Hip | Knee | Ankle | Mean |
|---|---|---|---|---|---|---|---|---|---|
| PoseTrack17 Val Set | Girdhar et al. [48] | 72.8 | 75.6 | 65.3 | 54.3 | 63.5 | 60.9 | 51.8 | 64.1 |
| | Xiu et al. [57] | 66.7 | 73.3 | 68.3 | 61.1 | 67.5 | 67.0 | 61.3 | 66.5 |
| | Bin et al [23] | 81.7 | 83.4 | 80.0 | 72.4 | 75.3 | 74.8 | 67.1 | 76.7 |
| | HRNet [27] | 82.1 | 83.6 | 80.4 | 73.3 | 75.5 | 75.3 | 68.5 | 77.3 |
| | MDPN [56] | 85.2 | 88.5 | 83.9 | 77.5 | 79.0 | 77.0 | 71.4 | 80.7 |
| | **PoseWarper** | 81.4 | 88.3 | 83.9 | 78.0 | 82.4 | 80.5 | 73.6 | **81.2** |
| PoseTrack17 Test Set | Girdhar et al. [48] | - | - | - | - | - | - | - | 59.6 |
| | Xiu et al. [57] | 64.9 | 67.5 | 65.0 | 59.0 | 62.5 | 62.8 | 57.9 | 63.0 |
| | Bin et al [23] | 80.1 | 80.2 | 76.9 | 71.5 | 72.5 | 72.4 | 65.7 | 74.6 |
| | HRNet [27] | 80.1 | 80.2 | 76.9 | 72.0 | 73.4 | 72.5 | 67.0 | 74.9 |
| | **PoseWarper** | 79.5 | 84.3 | 80.1 | 75.8 | 77.6 | 76.8 | 70.8 | **77.9** |
| PoseTrack18 Val Set | AlphaPose [58] | 63.9 | 78.7 | 77.4 | 71.0 | 73.7 | 73.0 | 69.7 | 71.9 |
| | MDPN [56] | 75.4 | 81.2 | 79.0 | 74.1 | 72.4 | 73.0 | 69.9 | 75.0 |
| | **PoseWarper** | 79.9 | 86.3 | 82.4 | 77.5 | 79.8 | 78.8 | 73.2 | **79.7** |
| PoseTrack18 Test Set | AlphaPose++ [56, 58] | - | - | - | 66.2 | - | - | 65.0 | 67.6 |
| | MDPN [56] | - | - | - | 74.5 | - | - | 69.0 | 76.4 |
| | **PoseWarper** | 78.9 | 84.4 | 80.9 | 76.8 | 75.6 | 77.5 | 71.8 | **78.0** |

As we vary the number of labeled frames per video (second leftmost figure), we notice several interesting patterns. First, we note that for a small number of labeled frames per video (i.e., $1-2$) our approach outperforms the "GT (1x)" baseline by a large margin. Second, we note that the performance of our approach and the "GT (7x)" becomes very similar as we add 2 or more labeled frames per video. These findings further strengthen our previous observation that PoseWarper allows us to reduce the annotation cost without a significant loss in performance.

### 4.3   Improved Pose Estimation via Temporal Pose Aggregation

In this subsection we assess the ability of PoseWarper to improve the accuracy of pose estimation at test time by using our deformable warping mechanism to aggregate pose information from nearby frames. We visualize our results in Figure 4 b), where we evaluate the effectiveness of our temporal pose aggregation during inference for models trained a) with a different number of labeled videos (second rightmost figure), and b) with a different number of manually-labeled frames per video (rightmost figure). We compare our approach ("GT (1x) + T-Agg.") to the same "GT (7x)" and "GT (1x)" baselines defined in the previous subsection. Note that our method in this case is trained exactly as "GT (1x)" baseline, the only difference comes from the inference procedure.

When the number of training videos and/or manually labeled frames is small, our approach provides a significant accuracy boost with respect to the "GT (1x)" baseline. However, once, we increase the number of labeled videos/frames, the gap between all three baselines shrinks, and the model becomes more saturated. Thus, our temporal pose aggregation scheme during inference is another effective way to maintain strong performance in a sparsely-labeled video setting.

### 4.4   Comparison to State-of-the-Art

We also test the effectiveness of our temporal pose aggregation scheme, when the model is trained on the full PoseTrack [22] dataset. Table 2 compares our method to the most recent approaches in this area [48, 57, 23, 27]. These results suggest that although we designed our method to improve pose estimation when training videos are sparsely-labeled, our temporal pose aggregation scheme applied at inference is also useful for models trained on densely-labeled videos. Our PoseWarper obtains 81.2 mAP and 77.9 mAP on PoseTrack2017 validation and test sets respectively, and 79.7 mAP and 78.0 mAP on PoseTrack2018 validation and test sets respectively, thus outperforming prior single frame baselines [48, 57, 23, 27].

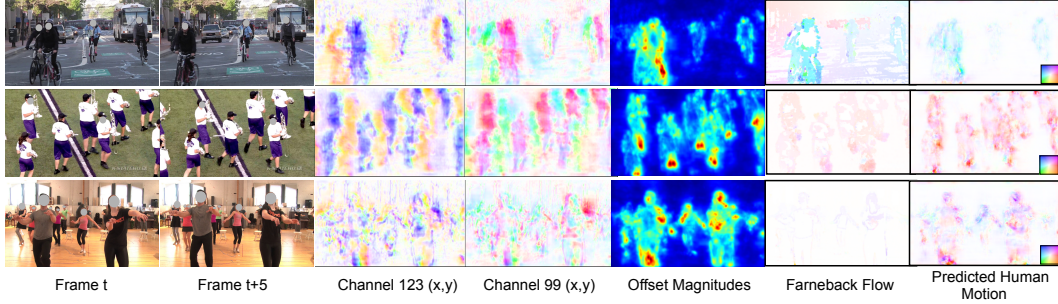

| Frame t | Frame t+5 | Channel 123 (x,y) | Channel 99 (x,y) | Offset Magnitudes | Farneback Flow | Predicted Human Motion |

Figure 5: In the first two columns, we show a pair of video frames used as input for our model. The $3^{rd}$ and $4^{th}$ columns depict 2 randomly selected offset channels visualized as a motion field. Different channels appear to capture the motion of different body parts. In the $5^{th}$ column, we display the offset magnitudes, which highlight salient human motion. Finally, the last two columns illustrate the standard Farneback flow, and the human motion predicted from our learned offsets. To predict human motion we train a **linear** classifier to regress the ground-truth $(x, y)$ displacement of each joint from the offset maps. The color wheel, at the bottom right corner encodes motion direction.

## 4.5 Interpreting Learned Offsets

Understanding what information is encoded in our learned offsets is nearly as difficult as analyzing any other CNN features [59, 60]. The main challenge comes from the high dimensionality of offsets: we are predicting $c \times k_h \times k_w$ $(x, y)$ displacements for every pixel for each of the five dilation rates $d$, where $c$ is the number of channels, and $k_h, k_w$ are the convolutional kernel height and width respectively.

In columns $3, 4$ of Figure 5, we visualize two randomly-selected offset channels as a motion field. Based on this figure, it appears that different offset maps encode different motions rather than all predicting the same solution (say, the optical flow between the two frames). This makes sense, as the network may decide to ignore motions of uninformative regions, and instead capture the motion of different human body parts in different offset maps (say, a hand as opposed to the head). We also note that the magnitudes of our learned offsets encode salient human motion (see Column 5 of Figure 5).

Lastly, to verify that our learned offsets encode human motion, for each point $p_n$ denoting a body joint, we extract our predicted offsets and train a *linear* classifier to regress the ground truth $(x, y)$ motion displacement of this body joint. In Column 7 of Figure 5, we visualize our predicted motion outputs for every pixel. We show Farneback's optical flow in Column 6. Note that in regions containing people, our predicted human motion matches Farneback optical flow. Furthermore, we point out that compared to the standard Farneback optical flow, our motion fields look less noisy.

## 5 Conclusions

In this work, we introduced PoseWarper, a novel architecture for pose detection in sparsely labeled videos. Our PoseWarper can be effectively used for multiple applications, including video pose propagation, and temporal pose aggregation. In these settings, we demonstrated that our approach reduces the need for densely labeled video data, while producing strong pose detection performance. Furthermore, our state-of-the-art results on PoseTrack2017 and PoseTrack2018 datasets demonstrate that our PoseWarper is useful even when the training videos are densely-labeled. Our future work involves improving our model ability to propagate labels and aggregate temporal information when the input frames are far away from each other. We are also interested in exploring self-supervised learning objectives for our task, which may further reduce the need of pose annotations in video. We will release our source code and our trained models upon publication of the paper.

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
