[Reviews · NeurIPS 2019]

Reviewer 1



- Here I will describe the proposed PoseWarper algorithm. -- The input consists of a pair of frames containing multiple humans and that are up to 3 frames apart. One frame, A contains 2D annotations of human body joints and frame B is unlabeled. The algorithm aims to predict poses for the frame B using only supervision from the frame A. First, a base network (using HRNet backbone [27]) predicts pose heatmaps for both frames. Then, their difference is computed and fed into a stack of residual layers, that predict per-channel offsets which are then used to warp pose heatmaps of the frame B. Finally, they compute the loss between the warped heatmaps and the ground truth heatmaps of the frame A. The warping mechanism is differentiable and is implemented in a similar fashion to the Spatial Transformer Networks by usin Deformable Convolutions [28]. This way the network learns a motion field that warps human body parts from between neighbor frames. -- The proposed methodarchitecture is validated on the task of propagation of poses by comparing against two baselines on the PoseTrack dataset [22]. The first baseline involves training a standard human pose estimator on the available frames and simply apply the trained detector on the unlabeled frames. It reaches the accuracy 79.3 mAP. The more advanced baseline uses a trained optical flow network (FlowNet2 [29]) to propagate annotations from the lableed to unlabeled frames and attains 83.8 mAP. The proposed PoseWarper reaches 88.7mAP, a substantial improvement over both baselines. -- In another set of experiments the propagated annotations were used to augment manual annotations for training a pose estimation network. The method comes close to training with full supervision on all available frames, and substantially outperforms a baseline when only one frame is available. -- Finally, spatio-temporal pose aggregation at inference time also improves over naive baseline. - I find the paper overall well-written. Introduction is very clear and provides good motivation for the main part. Related work is thorough and comprehensive. Here are some questions and concerns: - Missing ablations for the architecture of the offset predictors. Why 5 levels are chosen? This need to be studied in more detail. - It is not clear how exactly poses from neighbor frames are aggregated in the Spatiotemporal Pose Aggregation. Are the warped heatmaps simply averaged? - What remains unclear whether the offsets are used to warp feature maps or the final keypoint heatmaps? Lines 115-116 say that "Next, the predicted offsets are used to spatially rewarp the pose heatmap" which indicates that offsets are warping body joint heatmaps, however later in the section 4.5 in the Figure 5 offsets are shown for the channel 123, which is clearly outside of number of keypoints in the PoseTrack dataset (14 keypoints). So does this mean that the feature maps are warped instead? - The presense of analysis section (4.5) is in principle good, but I cannot make much of it. In particular Figure 5 is not illustrative at all. I would like to see a more informative comparison between using state-of-the-art optical flow (why FlowNet2 is not used there?) and the proposed method. Also channels 123 and 99 are used, but it's not clear at all what those correspond, as I already mentioned in the previous remark.

Reviewer 2



Pros: - The paper is well written and easy to read. - The proposed PoseWarper network is novel and makes sense. Even though similar ideas have been explored in the literature, e.g, for facial keypoints in [53], I still found the ideas presented in the paper novel for body pose estimation. - The effectiveness of the proposed approach is demonstrated for three different applications on the validation set of PoseTrack-2017 dataset. Cons: - For frames with multiple persons, the proposed approach requires track ids of the persons to know which two bounding boxes should be used to learn the PoseWarper. I understand that track-ids are significantly easier to annotate as compared to poses, but this should be clarified in the paper. Currently, I couldn't find any explanation for this. - It's not clear how person association b/w frames are obtained during testing? Are the poses tracked following the approach in [23] before performing pose aggregation in Table-2? Conclusion: Overall it is a good paper with sufficient technical contributions. Exploiting temporal information for improving pose estimation in multi-person setups has been quite a challenge and this paper proposes a good way of doing this. However, I request the authors to kindly address my concerns below during the rebuttal.

Reviewer 3



Originality The task being addressed is in fact quite novel and also well motivated. Acquiring dense pose annotations in video could be tedious and time-consuming. This provides a strong case for studying pose estimation from sparsely labeled videos. Quality The methodology is technically sound. Evaluation on PoseTrack2017 is compared with proper recent approaches. The results might be more complete if the paper could add some ablation study on the proposed network architecture. Clarity Overall the idea and results are clearly exposed. It could be helpful to add a more rigorous description on deformable convolutions (L119-121). Significance The paper demonstrates three important tasks with the proposed technique (1, 2, 3 in the last block), which shows the wide applicability of the technique. ---Post Rebuttal Feedback--- The rebuttal does not change my view, which has been positive initially.

[Author Response · NeurIPS 2019]

**Requested Additional Results.**

**PoseTrack17 Results on Test Set (R2).** We evaluate our model on the PoseTrack17 test set (using our spatiotemporal
pose aggregation scheme) and obtain 77.94 mAP, ranking first on the PoseTrack17 leaderboard. We will add these
results to our final paper. We will also update Table 2 with missing entries from the leaderboard. We will cite the
missing papers and add them to our related work discussion.

**PoseTrack18 Results (R2).** We also evaluate our model on the PoseTrack18 dataset. Our spatiotemporal pose
aggregation scheme yields 80.1 and 78.0 mAP on the PoseTrack18 validation and test sets, respectively, ranking first
among entries that use only PoseTrack and COCO data, and second overall. We will include these results as well.

**Ablation Study on Dilated Convolution (R1, R2, R3).** Here we study the effect of different levels of dilated
convolutions in our PoseWarper architecture. We evaluate all these variants on the task of video pose propagation. First,
we report that removing dilated convolution blocks from the original architecture reduces the accuracy from 88.7 mAP
to 87.2 mAP. We also note that a network with a single dilated convolution (using a dilation rate of 3) also yields 87.2
mAP. Adding a second dilated convolution level (using dilation rates of 3, 6) improves the accuracy to 88.0. Three
dilation levels (with dilation rates of 3, 6, 12) yield a mAP of 88.4 and four levels (dilation rates of 3, 6, 12, 18) give
a mAP of 88.6. A network with 5 dilated convolution levels (Fig. 2 in the original draft) yields 88.7 mAP. Adding
more dilated convolutions does not improve the performance further. Additionally, we also experimented with two
networks that use dilation rates of 1, 2, 3, 4, 5, and 4, 8, 16, 24, 32, and report that such models yield mAPs of 88.6 and
88.5, respectively, which are slightly lower. We will add this ablation study to our final paper.

**Requested Clarifications.**

**Spatiotemporal Pose Aggregation. (R1)** We chose to average the warped heatmaps from neighboring frames during
spatiotemporal pose aggregation, as we discovered it to be a simple yet effective scheme.

**Warping Heatmaps (R1).** The offsets are used to warp the pose heatmaps. We predict $c \times k_h \times k_w \times 2$ offset channels
(i.e., $(x, y)$ displacements) for every pixel where $c$ is the number of joints, and $k_h, k_w$ are the deformable convolution
kernel height and width respectively (see L271-273). In our case, $c = 17$, and $k_h = k_w = 3$, which means that we
predict 153 $(x, y)$ displacements (306 channels) for every pixel.

**Interpretability of Offsets and Comparison with FlowNet2 (R1).** We agree with R1 that it is difficult to understand
what our predicted offsets encode based on their direct visualizations. However, Figure 5 reveals that different offset
maps encode different motions and suggests that the method performs some sort of motion decomposition corresponding
to different body parts or discriminative regions. Figure A1 of this document includes a more intuitive illustration of
the motion encoded by PoseWarper and compares it to the optical flow computed by FlowNet2 (as requested by R1)
for the video pose propagation task. The first frame in each 3-frame sequence illustrates a *labeled* reference frame at
time t. For a cleaner visualization, we show only the "right ankle" body joint for one person, which is marked with a
**pink** circle in each of the frames (please zoom in). The second frame depicts our propagated "right ankle" detection
from the labeled frame in time t to the unlabeled frame in time t+1. The third frame shows the propagated detection
in frame t+1 produced by the FlowNet2 baseline. This visualization and similar ones that we generated and that we
plan to include in supplementary material, suggest that FlowNet2 struggles to accurately warp poses if 1) there is large
motion, 2) occlusions, or 3) blurriness. In contrast, our PoseWarper handles these cases robustly, which is also indicated
by our results in Table 1 of our submission (i.e., 88.7 vs 83.8 mAP w.r.t. FlowNet2). For the final version of our paper,
we will add more visualizations such as the ones in Figure A1. We believe that they will provide a better qualitative
understanding of why our method is advantageous compared to FlowNet2 (besides the quantitative benefits of lower
computational cost and improved accuracy, which we already discussed in our submission).

**Track ID Annotations (R2).** Our approach does *not* require track ID manual annotations or externally generated
person tracks. We train our model to warp pose heatmaps from an unlabeled Frame B to a labeled Frame A. For each
labeled bounding box of a person in Frame A, we crop Frame B at the same location using a bounding box that is large
enough to include the same person even if he/she moved. During spatiotemporal pose aggregation inference, we apply
the same scheme and use bounding boxes automatically extracted with a detector from [48] (see L163-165). We note
that because our predicted offsets encode motion cues between Frames A and B, we can potentially leverage our offsets
for tracking. Furthermore, as noted by R3, our proposed framework is general enough to be applied for other video
tasks such as object detection or instance segmentation in video.

Figure A1: Comparing our PoseWarper and FlowNet2 for video pose propagation. Please zoom in to see the **pink** circle on the right ankle. Unlike our model, FlowNet2 fails to accurately propagate poses when there is large motion, blurriness or occlusions.

[Meta-Review · NeurIPS 2019]

All reviewers recommended accept.